# Favorable Adaptation during SARS-CoV-2-Pandemic as Told by Student-Athletes—A Longitudinal and Behavioral Study

**DOI:** 10.3390/ijerph191912512

**Published:** 2022-09-30

**Authors:** Urban Johnson, Krister Hertting, Andreas Ivarsson, Eva-Carin Lindgren

**Affiliations:** School of Health and Welfare, Halmstad University, SE-30118 Halmstad, Sweden

**Keywords:** resilience, corona mental health, focus group interviews, youth athletes, biopsychosocial approach, SARS-CoV-2-pandemic

## Abstract

(1) Background: The aim was to explore the impact of adaptive responses (resilience factors) on student-athletes’ behaviors during a stressful period of life during the SARS-CoV-2-pandemic of 2020 and 2021. (2) Methods: A constructivist-based grounded theory (CGT) was used guided by a biopsychosocial approach. Based on purposeful sampling, 22 male and female student-athletes were divided into four focus group interviews (FGI) seven months after the outbreak of the SARS-CoV-2-pandemic (October 2020) and 18 of these were followed up by FGI seven months later (May 2021). The mean age was 17.65 and they represented golf and handball. (3) Results: The CGT contained four main categories seven months after the SARS-CoV-2-pandemic outbreak: Social support, Self-discipline, Physical practice and recovery and Management of everyday life. Fourteen months after the SARS-CoV-2-pandemic outbreak, four additional main categories emerged: More extensive social support, Belief in the future, Self-awareness and Responsibility and problem-solving. (4) Conclusions: The CTG suggests that student-athletes’ favorable adaptations to the stressful SARS-CoV-2-pandemic period 2020–2021 are based on a gradually developed ability to take responsibility for one’s own actions, insight into the importance of deepened social interaction, belief in a positive post-COVID future and increased awareness of physical activity and its relation to perceived health.

## 1. Introduction

The SARS-CoV-2-pandemic is a global crisis that can substantially negatively affect mental, physical and social aspects of people’s lives. The increased prevalence of mental health problems worldwide is one example [1,2,3]. Higher levels of mental health problems and lower levels of well-being have, for example, been found in different sub-populations, such as young people [4] and young athletes [5]. Studies have also shown that more elite athletes were at-risk for clinical levels of depression and COVID-specific worries [6] and worse mental and emotional health during the SARS-CoV-2-pandemic [7]. One potential explanation might be that athletes are groups who have experienced significant changes in their daily habits, rituals and routines, particularly during the initial lockdown period. These changes in everyday life might increase the risk for mental health problems and reduce well-being. One group particularly exposed to substantial changes is that of student-athletes. Both changes in training and school routines were, for many within this population, substantial stressors, especially during the beginning of the pandemic [8]. Research indicates [9], for instance, that social distancing and being away from teammates, were concerns raised among community college student-athletes as consequences of the pandemic. The changed form of training and school routines generated significant problems for some student-athletes. In contrast, it brought opportunities for developed self-responsibility and increased self-insight for others. In an interview study of student-athletes during the SARS-CoV-2-pandemic [10], it was shown that adaptive coping strategies, such as the experience of positive encouragement and social support from classmates and coaches, were common themes positively affecting their experiences returning to sport and campus life. Previous findings show that people and institutions might react very differently during extensive stressors (e.g., global crises) [11].

As mentioned, the SARS-CoV-2-pandemic brought health-related problems, but also possibilities for favorable adaptation for others. Focusing on favorable adaptation to the SARS-CoV-2-pandemic, studies have, for example, shown that high levels of physical activity [12], adaptive coping strategies such as keeping daily routines [13], as well as perceived support [14,15], were all related to improved mental health and well-being among both student-athletes and competitive athletes. One factor suggested to be essential to understand how persons successfully cope with major stressors is resilience. Several scholars have put forward different definitions of these concepts. A commonly used and accepted definition reads, “Resilience refers to a dynamic process encompassing positive adaptation within the context of significant adversity” [16], p. 543.

While most of the previously published studies on resilience and favorable adaptation in a sports context are characterized by a quantitative design, relatively few have adopted a qualitative design [11]. One such study is of particular interest because it focusses on resilience in a sports context and is based on a grounded theory, inspired by Strauss and Corbin, 1998 [17]. The results indicated that several psychological factors such as a positive personality, motivation, confidence, focus and perceived social support seemed to protect Olympic champion athletes from the potential negative effect of stressors by influencing their challenge appraisal and metacognition. These factors appeared to promote facilitative responses that preceded optimal sports performance. However, with few exceptions [11,18], a lack of longitudinal studies concerning resilience in a sport is also noted, especially concerning those with a qualitative design, which limits the possibility for researchers to detect developments or changes in the characteristics of the study population. As suggested by Galli and Vealy [19], qualitative studies on resilience should adopt a longitudinal interview schedule that not only would make recall bias less likely but would allow for a deeper examination of dynamic thoughts and processes during adverse situations.

A steadily increasing number of studies also support a multiple-level model of resilience towards various stressors, indicating the effects of biological and psychosocial factors among young people [20]. This kind of research acknowledges study limitations, including cross-sectional design when using a biopsychosocial approach to studying changes over time in resilience among adolescents [21]. A biopsychosocial approach requires a transdisciplinary perspective integrating the interplay of biological, psychological and social factors and is, not least, important in contexts where physical activity is centered. Thus, adopting a biopsychosocial perspective in the study of resilience in sport opens promising and multidisciplinary possibilities of gaining increased knowledge about the positive impact it has on athletes’ behavior, yet, to our knowledge, no study directed towards the student athlete’s experiences related to the SARS-CoV-2-pandemic is to be found.

To sum up, research has shown that the SARS-CoV-2-pandemic is associated with increased risk for mental health problems, such as COVID-specific worries and increased depressive symptoms, in young adults and athletes. Research has also presented mental health challenges and responses related to favorable adaptation to the SARS-CoV-2-pandemic, such as keeping daily routines and perceived support. Most published research papers on sport and resilience seem to be based on quantitative data, few adopt a longitudinal design and even fewer use a biopsychosocial perspective related to the SARS-CoV-2-pandemic among student-athletes. Based on a constructive grounded theory, results from such a study have the potential to capture unique and valuable knowledge about student-athletes’ responses and behavior to the SARS-CoV-2-pandemic over time and give rise to implications for research and practice aiming to strengthen the readiness of student-athletes for future challenges. Consequently, this study aimed to develop a grounded theory of how adaptive responses (resilience factors) influenced student-athlete behaviors’ during the SARS-CoV-2-pandemic of 2020 and 2021.

## 2. Materials and Methods

### 2.1. Research Design

Considering the decision to explore how young student-athletes adapted to the SARS-CoV-2-pandemic of 2020–2021 at a certified sport-oriented high school in Sweden (CSOHS), a constructivist-based grounded theory (CGT) was deemed an appropriate methodological choice for the current study. Charmaz’ [22] CGT was selected as congruent with the researcher’s philosophical perspective because CGT devalues the existence of objective reality and instead claims that realities are social constructions. Therefore, CGT has a subjective epistemology in that it assumes that researchers are not separate from the research and that meaning is co-created between the researcher and the participants. According to Charmaz [22], CGT is a beneficial methodology when there is little pre-existing theory on a specific process or when existing theories do not adequately capture the complexity of the process. Previous research based on the student-athlete experience of the SARS-CoV-2-pandemic is limited and CGT methodology can advance the literature by offering new theoretical insights and explanations, producing rich and diverse data with a particular focus on the student-athletes’ own experiences.

### 2.2. Sampling and Participants

A purposive sampling [23] procedure was utilized to ensure a highly experienced set of male and female students from both individual and team sports, in their second or last academic year, who had experienced the outbreak and subsequent period of the SARS-CoV-2-pandemic. Participants were recruited from one certified sports-oriented high school (CSOHS) that typically has student-athletes who match this profile. The research group informed the management, sports coaches and teachers at this CSOHS about the purpose and nature of the study. Four different sports represent the CSOHS: ice hockey, football, golf and handball, as well as gym & fitness. In the current study, a team and an individual sport were chosen as study objects, i.e., handball and golf, because these sports were well represented in terms of potential participation in the CSOHS. The sports coach and teachers initially recruited the participants, aiming to identify both male and female student-athletes who were participating in either golf or handball.

Above all, the questions stimulated student-athletes to reflect on what psychosocial features characterize those who adapted well to the SARS-CoV-2-pandemic. That is, the student-athletes were asked to openly reflect on characteristic features of resilient behaviour that they recognized being particularly of value related to the SARS-CoV-2-pandemic.

A repeated interaction procedure between data collection of focus group interviews (FGI’s), analyses and theoretical sampling was employed according to CGT, an iterative research process [24]. The study also adopted a longitudinal design, where the sampling and data collection process took place seven and 14 months after the SARS-CoV-2-pandemic outbreak. Previous research has also advocated longitudinal design for resilience research [21].

The mean age of the student-athletes was 17.65 years of age at the time of FGI 1 and all had attended the CSOHS during the academic year 2020–2021 (see Table 1). Student-athletes either studied on the Social Science program or the Economics program, which means three-years of full-time study and the same student-athletes participated in FGI 1 and 2.

The presentation of the study stated that the sports student’s participation was voluntary, that they could withdraw at any time and that any information provided would be treated confidentially. All the contacted student-athletes and representatives of school management agreed to participate in the study and the student-athletes provided both oral and written informed consent. The Swedish Ethical Review Authority approved an ethics application (Dir. 2020–03716) which means, among other things, that no informed consent from parents was needed because all participants were over 15 years.

### 2.3. Procedures

Focus group interviews were chosen in order to capture the social interaction when individuals create meaning together. The study consisted of participants who are considered a homogeneous group (student-athletes), a strategy that enhanced group dynamics and interactions and therefore provided richness in the data [22]. Three to six student-athletes attended each FGI session, ending up with a total of eight FGIs (four seven months and four 14 months after the outbreak, see Table 1). The student-athletes discussed their sporting and school experience during the changed situation that followed the SARS-CoV-2-pandemic. Focus, content and clarity in a first interview guide were piloted with the first FGI. While this pilot work did not change the core issues of the interview guide, follow-up probes were modified to allow a detailed discussion of these factors in the following three FGIs. The follow-up FGIs 14 months after the SARS-CoV-2-pandemic outbreak contained more core issues that tried to capture the continuing process since the previous FGIs.

The interview guide was constructed to stimulate responses regarding the potential stressors (surprising adversity) caused by the pandemic, which challenged the student-athletes’ values and meta-cognitive experiences. However, the questions were developed from the emerging data and the ongoing analysis. The following key questions in the interview guide were: (a) Would you be able to tell each other how you experienced opportunities to conduct physical training sustainably during the spring of 2020? (b) How have you handled the obstacles you encountered? (c) In your opinion, what opportunities have any changed forms of sport/training contributed to? Relevant sequel questions followed the key questions. The CGT analysis of the data from one interview often informs the following direction [22]. By making memos directly after the FGIs about what was said, along with theoretical notes on data and a critical discussion in the research group, the FGIs were developed. Significantly, the FGIs’ direction in spring 2021 became driven by emerging theory. For example, the student-athletes were asked, “Could you tell each other what lessons from your changed life situation you have taken with you and kept since the last time we met?”, “In your opinion, what do you think characterizes the student-athletes who handled the SARS-CoV-2-pandemic particularly favorably?”.

At the beginning of each FGI, student-athletes were told that all statements would remain within the group, that there were no right or wrong answers and that all experience and opinions were significant. The FGIs were conducted at the school in October 2020 and the student-athletes considered this a safe environment [25]. In May 2021, when the opportunity was again given for physical attendance at the sports high school, the FGI was once again conducted in a safe environment. The FGIs seven months after the outbreak lasted between 45–55 min. The FGIs 14 months after the outbreak lasted between 25–40 min. The follow-up FGIs were shorter than on the first occasion, as the conversations on the follow-up occasion become more focused on what had happened since the last FGI and driven by the emerging theory, as already mentioned.

All FGIs were audio-recorded with a smartphone and professionally transcribed verbatim, yielding 118 pages of single-spaced text. The first and second authors were moderators and observers in all FGIs and both had previous experience in conducting FGIs. The moderator’s task was to ask questions, ensuring that everyone was involved in the discussions and followed the topic. The observer’s role was to monitor what was said and ask follow-up questions.

### 2.4. Data Analysis

We analyzed FGI’s intertwined with collection using initial line-by-line coding, analytic reflexive memoing, focused coding to bring relationships and patterns between initial codes into subcategories and categories and theoretical sampling [26]. The first and second author led the analysis process. At the start of this study, we used sensitizing concepts for instance resilience and our different disciplinary perspectives (sport psychology, pedagogy and sport science) as tentative tools for developing our ideas about the processes that we define in our data. The data analysis began after the first FGI seven months after the SARS-CoV-2-pandemic outbreak and continued in an iterative process for a month. First, each transcript was read one or more times to develop a sense of the overall context of the data and to gain familiarity with the demographic characteristics and detailed information provided. The focus then shifted to initial coding, which was performed via an initial line-by-line open coding on the FGI’s transcripts to take a fresh look at what the data communicates. In this phase, we asked questions of the data, such as “What is happening”? and “What are the processes that take place?” Analytical reflexive memos were written throughout the research process and were used to note interpretations, conceptual connections and patterns as they became evident to the researchers throughout this process [24]. In memoing, the recorded ideas about codes and the relationships among them, (sub)categories and properties were constructed by constantly questioning data and making the connection between what was discovered in the data and what we knew and had experienced. Throughout this process, we also discussed our own perceptions of the data and how our pre-understanding may have influenced our perceptions. Subsequently, focused coding was performed to explore preliminary subcategories in a comparative and iterative process. The tentative subcategories were created by constantly comparing for similarities, variations and differences in data, codes and subcategories, known as a constant comparative method [22]. In this process, the subcategories and properties were constructed by constantly questioning data and connecting what was discovered in the data. Theoretical sampling was used to deepen the analysis and develop and refine the categories in the emerging theory [22]. Thus, we tested new ideas to see how these concepts fit the collected data by constantly checking and rechecking. To evaluate the credibility of the coding, we reviewed and critically discussed the codes, subcategories and the main category repetitively. We returned to the original text to validate the model, which led to adjustments such as redefining and clarifying of some of the main categories. Furthermore, Charmaz [27], p. 259, referred to sensitizing concepts as “those background ideas that inform the overall research problem”. The sensitizing concept of resilience has guided this research but has only been used as a ‘point of departure’ to develop ideas [22]. All quotations were translated from Swedish into English and adjusted for readability. The student-athletes are identified with a pseudonym, as either male (m) or female (f) and by their focus group (e.g., FGI 1). Data produced from the student-athletes 14 months after the SARS-CoV-2-pandemic outbreak provided some unique perspectives on previously identified concepts and relationships among categories; the research group judged that saturation of theoretical concepts was achieved at this time point [22].

## 3. Results and Discussion

The core category, A powerful metacognitive experience and constructive evaluation, summaries the process whereby biopsychosocial (BPS) resilience has the function to safeguard student-athletes from the potential adverse effect of SARS-CoV-2-pandemic stressors. The generated CGT contained four main categories seven months after the SARS-CoV-2-pandemic outbreak: (1) Physical practice and recovery, (2) Self-discipline, (3) Social support and (4) Management of everyday life. Additionally, the main categories 14 months after the SARS-CoV-2-pandemic outbreak also comprises four main categories: (1) More extensive social support, (2) Belief in the future, (3) Self-awareness and (4) Responsibility and problem-solving. All these main categories constitute the characteristics and dimensions of the core category. The processes of powerful metacognitive experience and constructive evaluation seven months and 14 months after the SARS-CoV-2-pandemic outbreak seem to promote facilitative responses that precede successful adaptation to the SARS-CoV-2-pandemic. The content of the four main categories generated seven months after the outbreak is progressively and qualitatively followed by more nuanced and future-oriented main categories 14 months after the outbreak (see Figure 1).

### 3.1. Setting the Scene

The lockdown period for Swedish high schools, including CSOHS, because of the SARS-CoV-2-pandemic began in March 2020 and lasted until August 2020. During the period of August 2020 and until the last week of October the same year, the compulsory school system, as well as most public sports arenas (indoor and outdoor facilities) were open for training and competitive activities. The second lockdown period began in the first week of November 2020 and lasted until roughly the middle of April 2021 in Sweden. Almost the same closures affected physical training facilities and above all for indoor sports such as handball. By and large, all competitive activity, e.g., matches, tournaments, competitions and championships were canceled during this period for all sports apart from adult elite sports in Sweden.

#### COVID 19 as a Stressor

The outbreak of the SARS-CoV-2-pandemic in March 2020 acted as a significant stressor for most people, not least in the face of the changes it brought in disruptions to daily routines and the potential risk for mental health problems that followed, such as restricted possibilities for physical activity. Many of the established protective factors of resilience such as perceived/tangible social support [17,28], sense of meaning/belonging [29] and motivational climate [30] were severely disrupted during the SARS-CoV-2-pandemic. One group particularly exposed to substantial changes was adolescent student-athletes. For many within this population, changes in training routines (reduced time dedicated to sport) and school routines (transition to distance education) were experienced as a substantial stressor during the SARS-CoV-2-pandemic. For example, one student-athlete expressed his life situation during the first seven months of school closure and sports as follows (m/FGI 3):


*My whole life was a roller coaster; everything went up and down most of the time related to the set restrictions; one moment, we got to meet 50 people, the other moment, we only got to meet 5. We could do some physical training; then we were told to stop all physical training. Everything went up and down all the time. You never really knew how to organize your physical training, set up the social network and ultimately your schoolwork.*


The stressful period that SARS-CoV-2-pandemic entailed led to an extraordinary need to constructively manage their new life situation, not least for those student-athletes who had moved a long way from home to get top-quality training and academic opportunities for further development. For these and several other students, the SARS-CoV-2-pandemic was perceived as a major stressor and as a potential breeding ground for mental and physical illness.

### 3.2. The Narratives of the Focus-Group Interviews

‘A powerful meta-cognitive experience and constructive evaluation’.

The core category of this grounded theory was based on a powerful meta-cognitive experience and constructive evaluation of stressors among student-athletes. Powerful metacognitive experiences refer to a person’s awareness and feelings elicited in a problem-solving situation (e.g., feelings of knowing). This form of metacognitive skill is believed to play a major role in many types of cognitive activity, such as oral communication of information, reading comprehension, attention and memory [31]. It seems that some student-athletes associated successful handling of the SARS-CoV-2-pandemic stressor with being aware of the importance of managing their emotions and behaviour in various problem-solving situations, or, as the following student-athletes put it (f/FGI 5):


*I got to know myself better and what I needed to be more efficient at, such as dealing with negative thoughts and feelings. Because some may need the whole lesson to write, but I kind of learned how to put it together rather fast.*


A central aspect of understanding the student-athletes’ manner of cognitively dealing with the challenging stressor that SARS-CoV-2-pandemic brought was understanding how they developed constructive evaluations of important meta-skills such as organizing and communicating knowledge, judging the quality of information, giving and receiving feedback and improving self-assessment skills. For instance, one student-athlete expressed that (m/FGI 8):


*I will use the new and independent study routines I have learned at my sports high school when I start studying at the university because I think it will be a lot of personal responsibility and a lot of writing. So, it was something new for me and probably developing in the long run.*


Like the above quote, several student-athletes pointed out the importance of taking responsibility for organising their academic learning and knowledge and making short- and long-term judgments about the quality of information, such as future academic studies and professional sports.

#### 3.2.1. March–October 2020 (the First Seven Months, i)

##### Physical Practice and Recovery

The first main category contains two sub-categories, *“Motivation for training” and “Give the body time to recover after a long period of exercise”.* Under the first category, student-athletes expressed that they experienced increased self-training (golfer), which enhanced motivation. Some student-athletes seemed to be more motivated towards practice. Regarding the second sub-category, it is striking how student-athletes talk about the lock-down period’s opportunity to rehabilitate a previously exhausted and tired body. For instance, one student-athlete expressed that (f/FGI 2):


*… a positive thing was that we, who were a bit injured by overtraining, could take a long break and, thus, were able to rehab and slowly come back to more complex physical training. So, we got a lot of rest time and we could be in better shape.*


Research also shows that, in professional sport and specifically during injury rehabilitation, pronounced core themes representing psychosocial factors that help players cope successfully with rehabilitation were identified as constructive communication and rich interaction with significant others and a strong belief in the importance and efficacy of one’s actions [32].

##### Self-Discipline

The second main category refer to self-discipline and independence and the two sub-categories “*Quickly established routines regarded as helpful*” and “*Take care of myself*”. As for the first sub-category, student-athletes pointed out that they gradually learned to plan the day and, as a result, this created routines and gave structure to both school and sport. For instance, one student-athlete said(m/FGI 4):


*I think those who are structured in sports and have learned routines could then take help from that and structure the school; I think that goes hand in hand.*


The above quotation parallels results from research focusing on elite athletes, showing that the ability to keep daily routines was one factor associated with a reduced risk of mental health problems during selected periods of the SARS-CoV-2-pandemic [13]. Student-athletes also pointed out that the characteristics of those who managed this first part of the SARS-CoV-2-pandemic period in an effective way also learned quite quickly to stand on their own two feet and take care of their actions, such as one student-athlete who reported (m/FGI2):


*… a lot of personal responsibility, because you had to learn what it is like to study at university because we were at home and studied all the time and had to plan our everyday life with the help of our own constructed schedule if there was to be any kind of organization.*


In addition, some student-athletes pointed out that individual athletes may have an advantage as greater personal responsibility is built into their training and competition logic.

##### Social Support

The third main category refers to social support and is based on the following sub-categories: “*Support from home*” and “*Support from close friends*”. Several student-athletes point out that unconditioned social support from home was invaluable in the critical and first period of SARS-CoV-2-pandemic, as well as support from close friends and coaches. However, several student-athletes lived far away from home without direct and physical contact with their family. Previous research also indicates that resilience and mental health maintenance during the SARS-CoV-2-pandemic is associated with a less controlling family environment [33]. For example, one student-athlete stated (f/FGI 3):


*Positive with the social network in the home to brainstorm thoughts & ideas without getting questioned. But also, the social support and encouragement from close friends inside and outside the sport were positive for healthy living and adaptation to the prevailing circumstances.*


##### Management of Everyday Life

The fourth main category partly parallels the second one regarding self-discipline. However, the focus is more on direct and practical management of the everyday situation, such as a constructive solution to emerging digital problems and the understanding that adaptation to the prevailing circumstances was an immediate and a necessary strategy for coping with school, training and social life. Two sub-categories can be distinguished, “Immediate management of everyday worries” and “*Creation of time for different activities simultaneously*.”

Regarding the first sub-category, some student-athletes mentioned the challenge of constructively dealing with the everyday situation that had arisen. A student-athlete said (m/FGI 1):


*We were pretty quickly forced to do things differently. This was good training for our ability to focus on the right things. Still, it also created many adjustment problems in the beginning because we no longer had a scheduled everyday life with training, school, training activities, etc.*


Other student-athletes talked about the difficult balancing act of getting the daily schedule together when the previously fixed activity times had more or less collapsed. A student-athlete put it this way (F/FG1 4):


*“Creating time for everything (school, training, social) was a challenge, but it got better over time”.*


#### 3.2.2. November 2020–May 2021 (the following Seven Months, ii)

##### More Extensive Social Support

During the following seven months of the SARS-CoV-2-pandemic, the student-athletes gave similar but more qualitatively in-depth answers to questions concerning constructive adjustment to prevailing circumstances. A typical main category concerns more extensive social support. This category comprises two sub-categories, *“In-depth and motivating dialogue with parents and other related persons regularly” and “Use available resources/networks, e.g., school psychologist”*. Regarding the first sub-category, some student-athletes expressed that they had lots of conversations with parents and other essential persons every day to keep their motivation up and not feel trapped at home. Other student-athletes also stated that they valued the family more than during the first SARS-CoV-2-pandemic. One of the student-athletes, in connection with the second sub-category, pointed out that (f/FGI 5):


*The school must have extra resources for a psychologist or counsellor aimed at students during COVID-19. This is especially important for sports schools where many students live away from home.*


Several researchers have pointed out the importance of both having solid social networks built on social and biological resources as protection to cumulative adversities [34], but also the ability to develop these networks during various forms of adversity to maintain mental health and well-being [7].

##### Belief in the Future

An additional and main tangible category aimed at adaptive adjustment to SARS-CoV-2-pandemic concerns a pronounced belief in the future. Two identified sub-categories form the basis of this category, “*Cognitive awareness and preparedness in case something should happen*” and “*Broadened the area of interest (within and outside sports)*”. Regarding the first sub-category, student-athletes believe that those who maintained a positive attitude to the stressful SARS-CoV-2-pandemic also set up several different and alternative plans for managing school and physical exercise. For this category of student-athletes, the dual commitment (school and sports) is an advantage, given that they work well together. A student-athlete expresses this (m/FGI 7):


*I think it is an advantage to have double-careers (school and sport) compared to non-sports students.*


It is also clear that some student-athletes have changed their sports investment to continue to keep their motivation up and thus prepare for a sports life after SARS-CoV-2-pandemic, in line with the second sub-category. One student-athlete said: I have changed my involvement in sports from golf to floorball. It is also interesting to note that several pointed out the adaptive advantage that they seem to experience as student-athletes have. Quotes are repeated several times such as (m/FGI 6):


*Student-athletes have not become sedentary. We always have different activities going on.*


##### Self-Awareness

During the latter part of the SARS-CoV-2-pandemic, student-athletes testified that the increased independence and responsibility for conducting progressive physical exercise also led to increased awareness of the benefits of physical exercise for the experience of growing health and well-being. This is manifested in the third main category, “*Self-awareness*”. This contains two subcategories, “Learned to know the body’s strengths and weaknesses” and “*Knowledge of the relationship that a healthy soul leads to a healthy body*”. To the first sub-category, “*Learned to know the body’s strengths and weaknesses*”, come statement such as (f/FGI 7):


*I have learned to know my body, so I gradually increased the training when the season opened again without being injured and overtrained. In this process, I read a lot on the internet about the gradual escalation of physical training and that I also asked my coaches about this.*


Student-athletes also said that they had learned in more detail how their body works and what they needed to eat to feel good and become physically stronger. Similar results have also been found by Shepherd et al. [35] pointing out that the balance between self-regulating physical training, food and digital school education has been a challenging and progressively accelerated stress for many student-athletes. In their study, those who had access to resources or maintained their physical activity throughout SARS-CoV-2-pandemic did not report changes to their overall health and wellness. Moreover, in a study [9] student-athletes commonly cited physical activity and their sport participation in particular, as a helpful outlet in response to the SARS-CoV-2-pandemic.

However, student-athletes expressed that they also learned to take advantage of lessons acquired during the SARS-CoV-2-pandemic, not least about how their physical status is related to the experience of well-being and to plan their everyday life appropriately. A student-athlete expresses it as (m/FGI 5):


*It is different now when I have to cook my food and keep my training schedule because in school there you got the schedule and then you ate at 11 after the training. But then I had to make my schedule. I learned a lot about how the body works and that to feel good, you have to look at the whole, I thought at least.*


##### Responsibility and Problem-Solving

The fourth and final main category concerns solving everyday problems and taking responsibility for schooling and physical exercise during the SARS-CoV-2-pandemic. The two concluding subcategories are “*Increased awareness of the importance of self-responsibility*” and “*Motivation to return to sports despite unclear circumstances*”.

A distinctive feature of student-athletes who experienced the SARS-CoV-2-pandemic favorably was that they took responsibility for their actions and showed an awareness that it was ultimately up to them if they would be able to take a double degree (academia and sports). One student-athlete expressed this as (f/FGI 7):


*I have learned self-discipline well before university, where there are not as many meetings with teachers.*


It is also notable that some student-athletes have developed clear problem-solving strategies that include constructive perseverance and motivation. For example, a student-athlete says that (f/FGI 5):


*Some are even more tagged to come back and show those who doubted one’s capacity, but then you also notice that others have gradually lost motivation because they are no longer at the top.*


### 3.3. Biopsychosocial Responses and Adaptation to the SARS-CoV-2-Pandemic

#### 3.3.1. Facilitative BPS Response

The processes of powerful meta-cognitive experiences and constructive evaluation seemed to promote a gradually developed facilitative response in student-athletes related to the SARS-CoV-2-pandemic from March 2020 to May 2021. In the first period of COVID (March–October 2020), student-athletes expressed constructive appraisals of the importance of having social contact and a support system to unburden the limited freedom that pandemics brought, in terms of schooling as well as if physical and collective training with peers, etc. Examples of this are expressed in the following excerpt (f//FGI 5):


*You have been able to ask other friends outside the close group, for example, “What do we have now or what should we do now”. So, you have been able to get help from each other often.*


When the student-athletes were in the second period of the SARS-CoV-2-pandemic, it was possible to note a qualitative change regarding the awareness and feelings elicited in problem-solving situations, for example, related to their use of social support and support systems inside and outside the home (e.g., in the form of informational, emotional and esteem-related support). Several student-athletes pointed out that they valued their family more now than before, especially in terms of deepened contact and more frequent communication with parents and siblings. They also emphasized the value of the collective training (team-athletes) that it was now gradually possible to participate in.

#### 3.3.2. Favorable Adaptation to the SARS-CoV-2-Pandemic

The student-athlete described patterns of favourable adaptation to the SARS-CoV-2-pandemic that helped them keep a satisfactory level of mental health and wellbeing during the period probably also led to the formation of constructive future plans for the post-COVID period to come. Gupta and McCarthy [36] discuss the concept of “sporting resilience” during the SARS-CoV-2-pandemic and how competitive elite athletes adapted to stressful situations. In their study, all participants engaged in positive reframing and challenge appraisal. The student-athletes also highlighted the importance of learning to use social support innovatively to overcome the various adversities faced. Learning and acting and gaining a sense of mastery in pandemic times also helped them successfully adapt and handle the SARS-CoV-2-pandemic. For instance, one student-athlete said that (m/FGI 7):


*… it is in some way self-discipline. After all, no one does it for you and no one comes and points out what you should do and not to do. If we were to succeed well, it was important to remember lessons learned, but at the same time, it was pretty tough because it came so immediately and without warning. The whole situation was involuntary for all of us.*


## 4. Main Discussion

Analysis of data suggests that student-athletes’ favourable adaptation to the stressful SARS-CoV-2-pandemic during 2020 and 2021 was based on a gradually developed ability to take responsibility for one’s actions, insight into the importance of deepened social interaction, belief in a positive post-COVID future and increased awareness of physical activity and its relation to perceived health. The results focus on factors that support student-athletes’ health and well-being and positive experiences of the pandemics; however, some narrative also indicates psychosocial factors that were experienced as troublesome. An in-depth discussion follows, focusing on results from the second data collection period and key concepts in Figure 1.

### 4.1. Development of Favorable Adaptation to the SARS-CoV-2-Pandemic over Time

The gradual development and progression of the various main and sub-categories from FGI 1 to FGI 2 indicate several forms of constructive adaptation to the prevailing stressful SARS-CoV-2-pandemic. It seems, for instance, that the student-athletes have increased their *self-awareness* that a reduction in their physical exercise and daily training routines also negatively influenced their ability to cope with everyday worries mentally. This result also parallels the concept of *biopsychosocial resilience* [21]. As described earlier, a steadily increasing number of studies support a multiple-level model of resilience towards various mental and physical stressors [37,38]. In the current study, the biopsychosocial context seemingly frames the student-athlete’s overall life situation, since very much of everyday life circulates around school and physical activity. The longitudinal design also provided an opportunity to understand the complex, but at the same time subtle interaction between the student-athletes’ desire to start training again with full intensity and their awareness of the need to “wait out” the end of the pandemic. This indicates a noticeable adaptation to a stressful life situation.

It is also visible in the stories from the student-athletes that favourable management of adversity is in many ways about having positive confidence in the future post-COVID period. This also includes the ability to be flexible and malleable regarding everyday life and, not least, the limited and in some cases, completely closed-off opportunity for physical exercise in indoor arenas and with peers. Examples of constructive *beliefs in the future* are adapting the previous sport’s profile to alternative sports and engaging in physical activities maintain motivation. In another study [10], some student-athletes also adopted new hobbies to help them cope with the stress related to the SARS-CoV-2-pandemic. The concept of adaptive coping, which is also mentioned as an indicator of successful management of adversities in sport in general [36,39] seems to be a relevant concept to use to describe the student-athletes’ adaptation to the current situation. In contrast, studies also show that, among competitive athletes at the SARS-CoV-2-pandemic outbreak, avoidant copers were characterized by the least effective coping skills [40].

As noted in the narratives, for most student-athletes at the end of their high school studies, this is a period in life involving powerful and robust psychosocial development. For many, a necessary form of identity and intimacy in both school and sport is developed. However, for some student-athletes, the SARS-CoV-2-pandemic might have triggered particular adaptive responses and perhaps a need for accelerated mental and physical maturity to manage the adversity flexibly and favorably. This maturation process includes psychosocial characteristics such as independent decision-making, evaluation, *responsibility and problem-solving* abilities, but also the ability to take advantage of an environment rich in personal, social, material and energy resources in order to provide safety and protection against resource loss and to promote resource growth. This line of reasoning coincides with parts of Erikson’s theory of lifelong psychosocial development through eight stages, which emphasis the social nature of human beings and the vital influence that social relationships have on growth [41]. According to Erikson, a conflict is a turning point when each person struggles to attain a specific psychological quality. This can be a time of vulnerability and strength as people work toward success or failure. The analysis results indicate that student-athletes associate turning points with opportunities for matured identity and a possibility for developed familiarity and extended social support. These qualities characterize a favourable transition to adulthood and parallel the core concept of resilience in several parts. Hobfoll [42] also underlines that those resources required for resiliency are acquired and aggregated across the lifespan and that persons in resource-rich environments are likely to accumulate resource gains.

Another favorable characteristic of gradual development during the SARS-CoV-2-pandemic is an increased ability in and awareness of maintaining and seeking *social help and support from* both the immediate family and more distant school and training peers. In another study [10], coaches were also widely seen as an important source of support during the SARS-CoV-2-pandemic, which parallels the results of this study. Interestingly, research conducted in the first phase of the SARS-CoV-2-pandemic indicates that student-athletes in their late teens, who received more social support and reported more connectedness with teammates, also reported better mental health and well-being [6]. It is not unlikely to assume that student-athletes who, already at an early stage of the SARS-CoV-2-pandemic, experienced strengthening social support, further developed this in the later period of the SARS-CoV-2-pandemic. Perhaps this also led to a stronger belief in one’s competence to handle adversity and, in the long run, a developed familiarity and strengthened identity, as in the previously discussed psychosocial development theory of Erikson. Findings [17] also suggest that perceived social support seemed to protect champion athletes from the potential adverse effect of experienced stressors.

### 4.2. Implications for Practice

Building on the presumption that future global crises and pandemics will strike society, it is of the utmost importance to take advantage of lessons learned from student-athletes’ experiences of how a successful adaptation to the SARS-CoV-2-pandemic can take place and how this can be used in practice for future preparedness. A holistic, grounded, systematic and educational approach to developing resilience for the future should preferably be endorsed at the group and societal level, but tailored to meet individuals’ needs and circumstances. The following two sets of implications for practice, based on the results of the FGI’s, are particularly important to highlight.

Improving the resilience of student-athletes with self-help strategies: student-athletes are encouraged to combine short-term goals and schedules and learn fundamental stress management techniques, as well as to continuously develop and maintain social support structures to successfully build resilience in daily life situations.

For sport high schools/sports clubs it is important to develop established peer support networks (peer systems): stimulated by support from school staff and, where applicable, with associated sports clubs, the development of peer support systems is likely to be necessary. Student-athletes are encouraged to create peer contacts and invite friends through network hubs or mentor support.

### 4.3. Limitations

In evaluating constructivist grounded theory research, it is important to consider its credibility, originality, resonance and usefulness [22]. To increase credibility, we conducted a rigorous methodological setting in accordance with the grounded theory methodology through the inductive-deductive cycle of theory generation. Credibility was also enhanced by the researcher’s knowledge of NIU and student-athletes and their interviewing of a range of participants in FGIs; the text was analysed by the authors who frequently discussed the interpretations and their pre-understanding and made systematic comparisons between FGIs, codes and categories. The new connections made among key themes from the psychological literature regarding resilience speak to the originality of this work. Originality was also enhanced through the implementation of follow-up FGIs which allowed us to analyze unique insights into student-athletes’ development of favourable meta-cognitive experiences and constructive evaluations to positively adapt to the SARS-CoV-2-pandemic. Similarly, the biopsychosocial perspective has opened a holistic understanding of the close connection between student-athletes’ two main core activities, school and physical training/sports, acknowledging the methodological advantages of using a biopsychosocial approach studying changes over time in resilience among adolescents based on student-athletes’ experiences of their training and school situation. Richly detailed descriptions and quotations are provided to enable readers to judge for themselves the trustworthiness of the data.

Resonance is highlighted by the presentation of multiple perspectives among the participants. Finally, based on our study design, it is problematic to comment on cause–effect relationships and generalize results to other contexts. It is possible that, if part of the study had been conducted in another country or a different cultural context, the variation could have been even richer. Our choice of a team sport and an individual sport could be expanded with additional and different teams and individual student-athletes, but we do not believe that this would have significantly increased the variation in the student-athletes’ stories. From FGI 1 to FG2 2, the drop-out rate was seven student-athletes (See Table 1). This may have affected the richness of the data during the analysis. However, the extensive stories we received at FGI 2 had probably increased the nuance and scope of the student-athletes’ narratives to a marginal extent. Moreover, a further limitation concerns the advantages and disadvantages of using focus groups compared to individual interviews. Limitations of focus groups include the tendency for certain types of socially acceptable opinion to emerge, the emphasis on dominant voices, and for some types of participants to dominate the research process [43,44]. However, the range of argumentative behaviors exhibited by participants may also result in a depth of dialogue not often found in individual interviews. The use of focus groups as a data-gathering method, concluding with spoken quotations to illustrate themes, ignores some of the complexities of focus group behaviour. Finally, the methodological challenges of selecting a particularly resilient group of student-athletes in the midst of the SARS-CoV-2-pandemic led us instead to ask a purposeful sample of student-athletes to reflect on resilient behavior in their current life situation. The results of this study must be interpreted from this starting point.

### 4.4. Future Research

Taking the insights from this study into consideration, several lines of future research are suggested. For instance, it is warranted to explore more closely the different constructs of the biopsychosocial model. Our study indicates that a holistic application of the BPS construct is motivated considering the student-athletes two dominant identities. Still, more specific measures of, for example, biological and/or physical markers such as salivary concentration of testosterone and cortisol might be useful. This has been successfully used in previously sports-related research related to psychological factors and based on the PBS framework [45]. This line of research might also be organized and implemented using a mixed-method research design [46], where all three areas are given equal space with appropriate measurements and data analysis and based on a longitudinal design. It would also be interesting to test our research findings compared to the results of studies of other global and extreme stressors/traumatic life events that might affect the health and well-being of student-athletes, e.g., financial crises/closure of schools, etc. Probably some of the results of this study are context specific. Still, it is likely that the results of this study will also generate complementing responses related to favourable adaptations of global stressors. This type of research can develop and add further nuances and knowledge to our grounded theory model of adaptation to the SARS-CoV-2-pandemic 2020–2021 and perhaps give rise to implications for practice to strengthen the readiness of student-athletes for future challenges.

## 5. Conclusions

This CGT study provides unique insights into student-athletes’ favourable adaptation to the stressful SARS-CoV-2-pandemic of 2020–2021. The results indicate a gradually developed ability to take responsibility for one’s actions, insight into the importance of deepened social interaction, belief in a positive post-COVID future and increased awareness of physical activity and its relation to perceived health.

## Figures and Tables

**Figure 1 ijerph-19-12512-f001:**
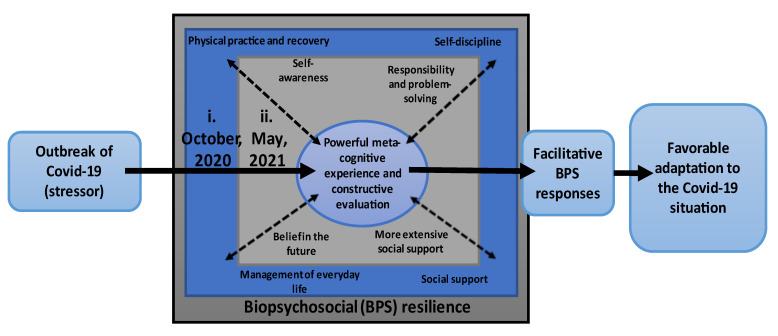
A longitudinal and schematic representation of the constructed main categories and the core category, illustrating their interrelationships in a grounded theory model of adaptation to SARS-CoV-2-pandemic 2020–2021 at a certified sport high school in Sweden. Answers belonging to i. correspond to March–October 2020 and answers belonging to ii. corresponds to the period November 2020–May 2021. Note: Parts of the construction of this figure are inspired by the previous published grounded theory of psychological resilience and optimal sports performance by Fletcher and Sarkar [17], p. 672.

**Table 1 ijerph-19-12512-t001:** Characteristics of the student-athletes.

Focus Groups	Sample (f/m)	Sports	Time-Interval
1	2 female/3 male	Golf, handball	7 months after the outbreak
2	2 female/4 male	Golf, handball	-//-
3	2 female/4 male	Golf, handball	-//-
4	1 female/4 male	Golf, handball	-//-
5	2 female/1 male	Golf, handball	14 months after the outbreak
6	2 female/4 male	Golf, handball	-//-
7	1 female/3 male	Golf, handball	-//-
8	1 female/3 male	Golf, handball	-//-

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
