# Peer review of "Favorable Adaptation during SARS-CoV-2-Pandemic as Told by Student-Athletes—A Longitudinal and Behavioral Study"

_ijerph, 2022, doi:10.3390/ijerph191912512_

Round 1

Reviewer 1 Report

Thank you for submitting the manuscript for review.  

The publication shows the favourable adaption from students’ athletes to the stressful SARS CoV-2 pandemic. Congratulations for this detailed and insightful research study. It sets a positive example alongside the wide range of negative stress and strain research conducted during the SARS Cov-2 pandemic. While reading the paper, a few thoughts occurred to me, which I would now like to present to you.

1)     I would recommend that instead of covid-19, you write SARS-cov-2-pandemic. Covid-19 is the disease of the infection with SARS-CoV-2 virus. After all, there are uninfected people in the pandemic. The title implies that ill student athletes were surveyed, but you were studying the process during the pandemic. You should correct it throughout the paper.

2)     Keyword: “constructivist-based grounded theory is part of the abstract. You should delete it. I suggest the keywords: interview. And add covid-19 (If you rephrase according to recommendation point 1).

3)     Introduction: The introduction is very detailed and understandable. Please put a point at the end of the sentence line 74.

4)     Material and methods: Please complete missing information according toConsolidated criteria for reporting qualitative research (COREQ): a 32-item checklist for interviews and focus groups” or equivalent.

Int J Qual Health Care 2007 Dec;19(6):349-57. doi: 10.1093/intqhc/mzm042

Consolidated criteria for reporting qualitative research (COREQ): a 32-item checklist for interviews and focus groups. Allison Tong, Peter Sainsbury, Jonathan Craig.

In particular, the questions of how the focus group was selected should be clarified. Were there any other inclusion or exclusion criteria?

5)     At what age is one considered to be of age in Sweden? Is a underage student required to have parental consent? Please complete.

6)     Line 185: What “memos” do you mean. Have you written down or recorded the conversations. In the proceeding, you write that interviews were transcribed. Specify exactly what equipment and software were used.

7)     Can you provide your interview guide as a supplement?

8)     The results, discussion, conclusions are well discussed. Overall, I also see it that sports students have high motivation anyway, e.g. training for professional sports or Olympics, etc. This can be reflected in the positive way they deal with the pandemic.

Author Response

Reviewers

Reviewer 1, 19 September 2022

Reviewer’s comments

Authors’ Responses

1)     I would recommend that instead of covid-19, you write SARS-cov-2-pandemic. Covid-19 is the disease of the infection with SARS-CoV-2 virus. After all, there are uninfected people in the pandemic. The title implies that ill student athletes were surveyed, but you were studying the process during the pandemic. You should correct it throughout the paper.

Thank you for addressing this correct comment. Accordingly, we have now replaced Covid-19 with SARS-cov-2-pandemic throughout the manuscript.

2)     Keyword: “constructivist-based grounded theory is part of the abstract. You should delete it. I suggest the keywords: interview. And add covid-19 (If you rephrase according to recommendation point 1).

We have deleted the keyword “constructivist-based grounded theory” and replaced this with the keyword “focus group interviews”. We have also added the keyword “SARS-cov-2-pandemic”.

3)     Introduction: The introduction is very detailed and understandable. Please put a point at the end of the sentence line 74.

A point is now added at the end of the sentence at line 74.

4)     Material and methods: Please complete missing information according to “Consolidated criteria for reporting qualitative research (COREQ): a 32-item checklist for interviews and focus groups” or equivalent.

Int J Qual Health Care 2007 Dec;19(6):349-57. doi: 10.1093/intqhc/mzm042

Consolidated criteria for reporting qualitative research (COREQ): a 32-item checklist for interviews and focus groups. Allison Tong, Peter Sainsbury, Jonathan Craig.

In particular, the questions of how the focus group was selected should be clarified. Were there any other inclusion or exclusion criteria?

We have now clarified the choice of using focus groups. We have formulated it as (line 170-173):

Focus group interviews was chosen in order to catch the social interaction when individuals create meaning together. The study consisted of participants who are considered a homogeneous group (student-athletes), a strategy that enhanced group dynamics and interactions and therefore provided richness to the data.

Thank you for the article recommended (Tong et al., 2007). The recommended article is included in the discussion (limitations).

Tong, A; Sainsbury, P; Craig, J. Consolidated criteria for reporting qualitative research (COREQ): a 32-item checklist for interviews and focus groups. Int. J. Qual. Health Care 2007, 19(6), 349-57. doi.10.1093/intqhc/mzm042

5)     At what age is one considered to be of age in Sweden? Is a underage student required to have parental consent? Please complete.

In Sweden, parental consent is required for children/young people under the age of 15. In our case this is not needed as the average age was 17.6 years. This information has been validated by the Swedish Ethical Review Authority approving our ethics application (Dir. 2020–03716).

We have clarified this information in manuscript line 167-168. Now it says: “… which means, among other things, that no informed consent from 15 years and older were needed”.

6)     Line 185: What “memos” do you mean. Have you written down or recorded the conversations. In the proceeding, you write that interviews were transcribed. Specify exactly what equipment and software were used

We have now written this clarification on line 233-236. Now it reads: In memoing, the recorded ideas about codes and the relationships among them, (sub)categories, and properties were constructed by constantly questioning data and making the connection between what was discovered in the data and what we knew and had experienced.

The conversation was recorded using smartphones. We have added this information at line 210.

7)     Can you provide your interview guide as a supplement?

Good comment! However, the basic questions are already included in the method section (procedures).  We hope this text answers your stated comment.

Reviewer 2 Report

It is with great pleasure that I reviewed the article entitled "Favorable Adaptation During Covid-19 as Told by Student-Athletes. A Longitudinal and Behavioral Study", whose purpose is to explore the impact of adaptive responses on student-athletes' behaviors during the pandemic period. 

To conduct this study, the authors used a qualitative method based on grounded theory with a constructivist approach. The study presents a panel of 22 students met in focus groups. 

Overall, the article is of good quality, the introduction contextualizes the issues very well.

However, the methodological aspects need to be strengthened. For example, if we know that the study took place in Sweden, it is necessary to specify the social context of the country. Indeed, Sweden has not developed a policy similar to the UK, USA, Canada or France in terms of student lockdowns. These elements appear insufficiently in the "Sampling and participants" section, line 136-137, and then we find elements in the last paragraph of section 2.3 "procedure". Was there a lockdown before the first interview in October? A better description of this situation would be a plus for the article. 

Next, the authors present the Data Analyses section, which is precise and allows for a clear understanding of the authors' approach.

Then, the authors present a result/discussion part of great quality and which allows to put forward results with rigor.

We would like to congratulate the authors for this work.

In the limitation section, it would have been interesting to discuss the limitations of focus groups compared to individual interviews. Indeed, some groups had interviews lasting 25 minutes, which is not very long for a focus group interview. Were there any group effects that prevented some athletes from responding?

Author Response

Reviewer 2, 19 September 2022

Reviewer’s comments

Authors’ Responses

Overall, the article is of good quality, the introduction contextualizes the issues very well.

However, the methodological aspects need to be strengthened. For example, if we know that the study took place in Sweden, it is necessary to specify the social context of the country. Indeed, Sweden has not developed a policy similar to the UK, USA, Canada or France in terms of student lockdowns. These elements appear insufficiently in the "Sampling and participants" section, line 136-137, and then we find elements in the last paragraph of section 2.3 "procedure". Was there a lockdown before the first interview in October? A better description of this situation would be a plus for the article.

Thank you for the nice judgment regarding the introduction.

Under the sub-heading “3.1. Setting the scene” (Results and Discussion) we write about which periods the SARS-cov-2 pandemic affected Sweden in 2020-2021 and its limitations with a focus on school education and training opportunities for athletes. If you judge that this information is not sufficient, or if the information should be moved to the method section, we are positive to develop/move the information in accordance with your comment, for instance in relation to the last paragraph of section 2.3.

Next, the authors present the Data Analyses section, which is precise and allows for a clear understanding of the authors' approach.

Thank you for your comment.

Then, the authors present a result/discussion part of great quality and which allows to put forward results with rigor. We would like to congratulate the authors for this work.

Thank you very much for your appreciated comment. We have spent extensive time on these two parts to maintain a qualitatively high level.

In the limitation section, it would have been interesting to discuss the limitations of focus groups compared to individual interviews. Indeed, some groups had interviews lasting 25 minutes, which is not very long for a focus group interview. Were there any group effects that prevented some athletes from responding?

We have added four sentences in the limitation section which problematizes the use of focus groups compared to individual interviews and relevant references. Now it reads (lines 653-660):

Moreover, a further limitation concerns the advantages and disadvantages of using focus groups compared to individual interviews. Limitations of focus groups include the tendency for certain types of socially acceptable opinion to emerge, dominant voices, and for some types of participants to dominate the research process [43-44]. However, the range of argumentative behaviours exhibited by participants may also result in a depth of dialogue not often found in individual interviews. The use of focus groups as a data-gathering method, ending up with talking quotations to illustrate themes, ignores some of the complexities of focus group behaviour.

43. Smithson, J. Using and analysing focus groups: Limitations and possibilities. Int. J. Soc. Res. Methodol. 2000, 3(2), 103-119. doi:

10.1080/136455700405172

44. Tong, A; Sainsbury, P; Craig, J. Consolidated criteria for reporting qualitative research (COREQ): a 32-item checklist for interviews and focus groups. Int. J. Qual. Health Care 2007, 19(6), 349-57. doi.10.1093/intqhc/mzm042

Reviewer 3 Report

See attached

Author Response

Reviewer 3, 19 September 2022

Reviewer’s comments

Authors’ Responses

It is true that the young population presents higher rates of anxiety and stress in the pandemic situation, however the conclusion that states that in young athletes it could be due to lost routines and habits, conflicts with other studies that affirm that the practice of sports activity precisely provides a plus of resilience to overcome the difficulties that the non-athletic population does not have to the same extent, which means that in stress and psychological discomfort the population of sports practitioners obtain better results (López-Gutiérrez et al , 2021: Psychological discomfort and stress during confinement due to the covid-19 pandemic.

We agree that you can find some contradictory results concerning response among young athletes and senior (adults) athletes related to favorable adaptation to the SARS-cov-2-pandemic. In our study we have focus on finding support in the literature about favorable behavior in relation to the pandemics among student-athletes such as keeping daily routines and perceived social support from close friends and family.

Our hope is that this is apparent from our literature search, but, as previously mentioned, we are aware of the ever-increasing proportion of published studies around athletes' experiences of SARS-cov-2-pandemic and the multifaceted publication of empirical articles that is now accessible.

Comparative study between athletes and non-athletes). In this sense, we fully agree that studies have shown, for example, that high levels of physical activity, adaptive coping strategies such as maintaining daily routines, as well as perceived support, were all related to better mental health. and wellness among both student-athletes and competitive athletes. The introduction presents an interesting perspective of the situation causing the problem, reviews some of the most relevant studies in relation to the way in which the pandemic has affected the population of athletes, although some are missing that contradict or complement some of the statements that are provided. The longitudinal and qualitative perspective of the proposed object of study is considered very interesting.

Thank you for your overwhelmingly positive rating and comment on our introduction and methodology.

Taking into account that the specific theme of the research is based on

constructivist theory and being a qualitative study, the bibliography is

relatively adequate. However, the one used is relevant and helps to

understand and justify the context.

Thank you for your overall positive comment. We have added two more relevant references that deals with methodological issues concerning the use of a qualitative design. This relates especially to contextual issues when conducting interviews and focus groups, as well as advantages and disadvantages of using focus groups compared to individual interviews.

References:

43. Smithson, J. Using and analysing focus groups: Limitations and possibilities. Int. J. Soc. Res. Methodol. 2000, 3(2), 103-119. doi:

10.1080/136455700405172

44. Tong, A; Sainsbury, P; Craig, J. Consolidated criteria for reporting qualitative research (COREQ): a 32-item checklist for interviews and focus groups. Int. J. Qual. Health Care 2007, 19(6), 349-57. doi.10.1093/intqhc/mzm042